# In-situ Adaptation for LLM-based Link Prediction: A Dynamic Cognition Paradigm for Temporal Knowledge Graphs

## Abstract

Knowledge graphs continuously evolve, making the prediction of future links a crucial but challenging task. Current methods, whether based on dynamic graph neural networks or static retrieval-augmented generation (RAG) for Large Language Models (LLMs), struggle with generalization. They often fail to capture the real-time, characteristic evolution of the graph during the testing phase, leading to degraded performance from distribution shifts. To address this, we propose a new training-free paradigm, termed **Dynamic Cognition** (**DyCo**), which posits that effective link prediction hinges on an agent's ability to continuously perceive graph evolution and adapt its strategies in-situ. Inspired by this, we introduce a novel framework **DyCo-LLM**, which enables an LLM to perform live adaptation for temporal link prediction. At its core is a dynamic context engine that tailors the LLM's prompts on the fly. This engine features an adaptive multi-path recall and scoring mechanism that adjusts its parameters based on the evolving node- and graph-level features. Furthermore, the framework incorporates a dynamic few-shot learner that generates corrective reasoning examples from prediction failures, allowing the LLM to learn from its mistakes in real-time without retraining. Experimental results on two large-scale dynamic knowledge graphs demonstrate that our approach achieves state-of-the-art performance in the link prediction task. Ablations verify that each recall path is indispensable, and balanced weights are critical to fuse structural–semantic signals and history–self similarity. In addition, the reflective few-shot routine provides consistent gains. The source code is available at `https://anonymous.4open.science/r/13htrueiwbgjkdsb/`.

## 1 Introduction

A Temporal Knowledge Graph (TKG) is a dynamically evolving structured representation of knowledge, where facts are encoded as timestamped edges between entities. A long line of parameter-trained models formulates TKG forecasting as autoregressive or rule/path-based reasoning with learned representations—e.g., recurrent or sequence models for future events (RE-NET), explainable subgraph expansion (xERTE), and temporal logical rules (TLogic) (Jin et al., 2019; Han et al., 2020; Liu et al., 2022). These approaches achieve solid performance but typically require re-training or careful fine-tuning to track distribution shift, can overfit to dataset idiosyncrasies, and often struggle with inductive generalization or realistic evaluation under harder negatives (Poursafaei et al., 2022; Gastinger et al., 2024).

In parallel, a second line of work enhances LLMs with specialized memory mechanisms for TKG forecasting: using in-context learning directly for TKG (ICL, see Fig. 1), combining rule-guided temporal retrieval with few-shot instruction tuning (GenTKG), or constructing historical contexts via analogical replay (AnRe) (Lee et al., 2023; Liao et al., 2024; Tang et al., 2025). This direction shows promising zero-/few-shot generalization (Zhao et al., 2023; Brown et al., 2020), but exposes a deeper bottleneck: memory and retrieval remain static and weakly coupled to the graph's live dynamics—fixed windows or hierarchical paging cannot expand/update with event accumulation, long-context recall amplifies noise (Packer et al., 2023; Jiang et al., 2024; Wu et al., 2025), and write/eviction is not conditioned on query-time structure, semantics, or recency—leading to failures

under tight input budgets and motivating an in-situ, diagnosis-driven adaptation of recall, scoring, and prompting.

To address these challenges, we argue that effective temporal knowledge graph reasoning must account for two crucial aspects of dynamic evolution: the continuous accumulation of event data, which demands a memory mechanism capable of efficient expansion and updates, and the shifting relevance of historical information as the graph evolves, which necessitates intelligent selection of the most pertinent context under limited input constraints.

Motivated by these requirements, we introduce **DyCo-LLM**. DyCo-LLM treats the LLM as a self-regulating reasoner that adapts at inference time through a closed loop: **Adaptive Parameters** first diagnose the current query's structural neighborhood, semantic associations, and temporal recency, and produce a set of runtime parameters that steer subsequent steps; **Multi-path Recall Process** then retrieves candidates along structural, semantic, and temporal paths in parallel, with the diagnostic outputs adaptively governing recall strength and complementarity; **Dynamic Score Calculation** fuses multi-view

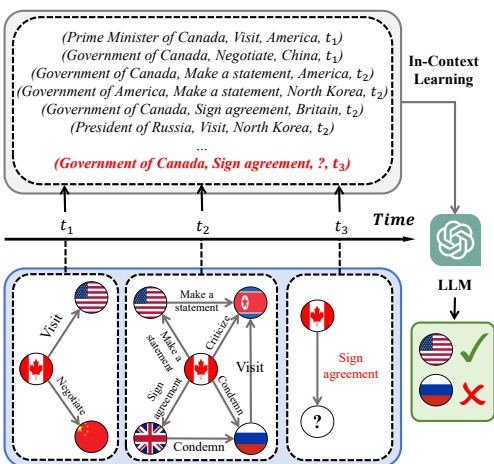

Figure 1: Convert the temporal knowledge graph into a textual structural context and use LLMs for link prediction.

evidence together with history/self-similarity trade-offs to rank candidates and set thresholds, yielding a compact, targeted context for the LLM; **Self-diagnosis Reasoning** analyzes failure cases, synthesizes targeted few-shot exemplars, and feeds them into subsequent, similar queries to achieve continual improvement. This closed loop enables the model to adapt in-situ without updating weights.

We conduct comprehensive evaluations on two standard TKG benchmarks (ICEWS and GDELT) (Garcia-Duran et al., 2018; Leetaru & Schrodt, 2013). Baselines span both routes: parameter-trained (JODIE, TGAT, DyGFormer, etc.) (Kumar et al., 2019; Trivedi et al., 2018; Xu et al., 2020; Wang et al., 2022; 2021; Cong et al., 2023; Yu et al., 2023) and LLM-based (ICL, GAD, AnRe) (Lee et al., 2023; Lei et al., 2025; Tang et al., 2025). Our main results use a Qwen3 (Yang et al., 2025) as the core reasoner. The experimental results indicate that DyCo-LLM consistently surpasses representative LLM-based methods and remains competitive with parameter-trained models, achieving SOTA results under the inductive setting and demonstrates excellent generalization ability. Ablations demonstrate that all three recall paths and the dynamic scoring are indispensable, while the adaptive parameters exhibit stable interpretability.

Our contributions can be summarized as follows:

- We propose a **training-free**, **test-time adaptive** framework. DyCo-LLM performs query-level adaptation via runtime diagnosis that jointly drives multi-path recall, dynamic scoring, and reflective prompting—achieving in-situ adaptation without updating weights.
- We unify the paradigms of dynamic graph representation learning and large language model memory research for knowledge graphs, proposing a **unified model framework**.
- Our method achieves **state-of-the-art** performance on knowledge graph link prediction tasks without requiring any fine-tuning, even surpassing most approaches trained on dedicated datasets.

## 2 PRELIMINARIES

**Temporal Knowledge Graph.** A Temporal Knowledge Graph (TKG) is represented as a sequence of timestamped facts, denoted as $\mathcal{G} = \{e_1, e_2, \ldots, e_N\}$, where each fact $e_k$ is a quadruple $(u, r, v, t)$. Here, $u, v \in \mathcal{E}$ are the head and tail entities from a set of entities $\mathcal{E}$, $r \in \mathcal{R}$ is the relation from a set of relations $\mathcal{R}$, and $t \in \mathcal{T}$ is the timestamp from a set of discrete timestamps $\mathcal{T}$, indicating when the interaction occurred. The facts are chronologically ordered, such that for any $k < j$, the

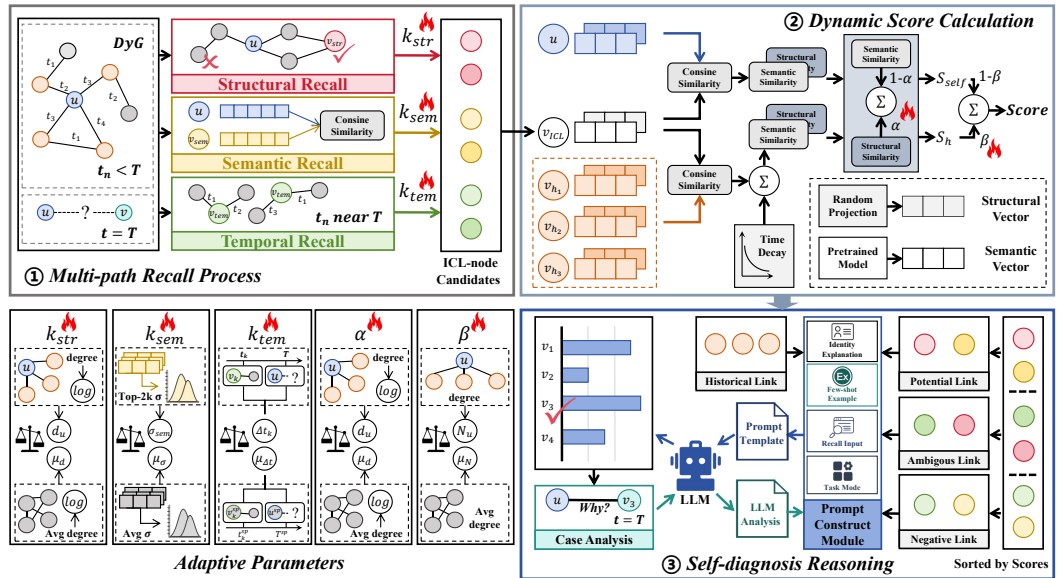

Figure 2: Overview of DyCo-LLM. (1) **Multi-path Recall Process:** Retrieves a tailored candidate set from multiple dimensions using the adaptive k-values. (2) **Dynamic Score Calculation:** Scores and partitions candidates using adaptive weights to construct a context-rich prompt. (3) **Self-diagnosis Reasoning:** Generates explanatory few-shot examples from incorrect predictions to dynamically refine the model's knowledge. (4) **Adaptive Parameters Module:** Generates in-situ hyperparameters by diagnosing the query's local context.

corresponding timestamps satisfy $t_k \leq t_j$. The TKG can also be viewed as a stream of subgraphs $\mathcal{G} = \{\mathcal{G}_1, \mathcal{G}_2, \ldots, \mathcal{G}_{|\mathcal{T}|}\}$, where each $\mathcal{G}_t$ contains all facts that occurred at timestamp $t$.

**Link Prediction on TKG.** The task of link prediction on a TKG is to forecast future interactions based on historical facts. Given the graph $\mathcal{G}_{<t_q} = \{(u, r, v, t) \in \mathcal{G} | t < t_q\}$ containing all interactions before a query time $t_q$, the goal is to predict the missing tail entity for a query $(u_q, r_q, ?, t_q)$. The model is tasked with ranking all candidate entities $v \in \mathcal{E}$ based on their likelihood of completing the quadruple, and the primary objective is to assign the highest rank to the ground-truth entity $v_q$.

**In-context Learning.** In-context Learning (ICL) is an emergent capability of Large Language Models (LLMs) to perform tasks by conditioning on a structured textual prompt, without any updates to the model's parameters. A prompt $\mathcal{P}$ is constructed to provide the LLM with both task-specific context and the query. Typically, $\mathcal{P}$ consists of a set of demonstrations (few-shot examples) $\mathcal{C} = \{(x_1, y_1), \ldots, (x_k, y_k)\}$ and a query $x_q$. The model $\mathcal{M}$ is expected to generate the answer $y_q$ by inferring the underlying task from the examples, i.e., $y_q = \mathcal{M}(\mathcal{C}, x_q)$.

## 3 METHODOLOGY

**Principle.** We pursue *in-situ adaptation*, treating the LLM as a cognitive agent that *diagnoses* each query's local context and adapts its prediction strategy accordingly. Concretely, the system *dynamically* generates a small set of hyperparameters that govern candidate recall, evidence fusion, and prompt-based reasoning, allowing the model to shift focus between structural and semantic signals as needed.

**Framework.** As shown in Figure 2, the **Adaptive Parameters** module inspects the query node $u$ and produces five hyperparameters $(k_{str}, k_{sem}, k_{time}, \alpha, \beta)$, which steer the downstream pipeline: (i) **Multi-path Recall Process** retrieves a tailored candidate set using the adaptive $k$'s; (ii) **Dynamic Score Calculation** ranks and partitions candidates under the adaptive weights $\alpha$ and $\beta$, yielding a compact, informative prompt for the LLM; and (iii) **Self-diagnosis Reasoning** integrates error

reflection dynamically into the reasoning process as few-shot examples to enhance subsequent queries. Detailed components appear in §3.1, §3.2, §3.3, and §3.4.

## 3.1 Multi-path Recall Process

To avoid prohibitive full-graph scans on large temporal KGs, we retrieve a compact, high-quality candidate set $C$ for query $(u, r, ?, t)$ via three complementary strategies that reflect distinct facets of graph evolution: (i) *temporal activity*, (ii) *structural proximity*, and (iii) *semantic similarity*. This multi-view design yields candidates that are both comprehensive and context-aware. Formally, given $(u, r, ?, t)$, the module retrieves three subsets:

**Structural Recall ($C_{\text{struc}}$)** captures local graph topology and community structure. Specifically, we retrieves the $k_{\text{struc}}$ nodes with the highest number of common neighbors with $u$, based on the graph adjacency structure up to time $t$.

**Semantic Recall ($C_{\text{sem}}$)** selects the $k_{\text{sem}}$ nodes whose textual descriptions (e.g., entity names or attributes) are most semantically similar to that of $u$. We use a pre-trained sentence transformer to encode entity texts and compute cosine similarities in the embedding space.

**Temporal Recall ($C_{\text{time}}$)** identifies the $k_{\text{time}}$ most recently active nodes in the graph prior to time $t$. This is implemented by scanning historical events in reverse chronological order and selecting nodes that interacted closest to $t$.

The final candidate set is the union of these three subsets: $C = C_{\text{time}} \cup C_{\text{struc}} \cup C_{\text{sem}}$.

## 3.2 Dynamic Score Calculation

Given the candidate set $C$ from multi-path recall, this module assigns each $v \in C$ a scalar score $S_{\text{final}}(v)$ indicating the likelihood of forming a future link with $u$ at time $t$. The scoring follows a *weighted fusion* of two complementary signals: (i) similarity to $u$'s **historical partners** (historical similarity) and (ii) **self-similarity** between $u$ and $v$. This dual-view design leverages past interaction patterns while preserving direct feature affinity. Algorithm 1 summarizes the procedure, which proceeds as follows:

**Step 1: Historical partner weights.** Frequent and recent partners of $u$ are stronger behavioral priors, we therefore emphasize them when aggregating evidence: $W(p) = \text{Count}(u, p) \cdot \exp\big( - \lambda_{\text{time}} (t - t_{\text{last}}(u, p)) \big)$, where $\text{Count}(u, p)$ is the number of past interactions between $u$ and partner $p$ before $t$; $t_{\text{last}}(u, p)$ is their last interaction time; $\lambda_{\text{time}} > 0$ is a fixed temporal decay rate; $t$ is the query time.

**Step 2: Similarities.** Structure captures network topology while semantics captures textual/attribute cues. We introduce a weighted blend reduces single-view bias:

$$M_{\text{total}}(v, p) = \alpha \, Sim_{\text{struc}}(v, p) + (1 - \alpha) \, Sim_{\text{sem}}(v, p),$$

where $Sim_{\text{struc}}$ and $Sim_{\text{sem}}$ are cosine similarities in structural and semantic embedding spaces, respectively; $\alpha \in [0, 1]$ adaptively balances structural vs. semantic evidence; $\mathbf{e}_{\text{struc}}(\cdot)$ comes from the dynamic random projection module (see Appendix E), and $\mathbf{e}_{\text{sem}}(\cdot)$ from a pre-trained sentence/attribute encoder.

**Step 3: Historical similarity.** Candidates resembling *weighted* historical partners are more plausible future links, reflecting repeated interaction patterns. Thus, we define the formula as follows:

$$S_{\text{hist}}(v) = \sum_{p \in \mathcal{P}(u)} M_{\text{total}}(v, p) \, W(p),$$

where $\mathcal{P}(u)$ is the set of $u$'s partners before $t$; $M_{\text{total}}(v, p)$ and $W(p)$ are from Steps 1–2.

**Step 4: Self-similarity.** The self-similarity score directly measures the affinity between the candidate $v$ and the query node $u$ itself, again in both structural and semantic spaces: $S_{\text{self}}(v) = \alpha \cdot Sim_{\text{struc}}(v, u) + (1 - \alpha) \cdot Sim_{\text{sem}}(v, u)$.

**Step 5: Final score and partition.** The final score for candidate $v$ is a weighted average of the historical and self-similarity scores, controlled by the adaptive parameter $\beta$: $S_{\text{final}}(v) = \beta \cdot S_{\text{hist}}(v) + (1 - \beta) \cdot S_{\text{self}}(v)$. We sort candidates by $S_{\text{final}}$ and partition them into *positives*, *ambiguous*, and *negatives* using thresholds $(p, q)$.

## 3.3 SELF-DIAGNOSIS REASONING

This module links statistical signals from the graph to the LLM's symbolic reasoning by turning link prediction into *structured* evidence-based inference. It builds an informative prompt from the outputs of *Dynamic Score Calculation* and performs *dynamic few-shot learning* from errors, enabling real-time, training-free improvement (in-situ adaptation).

**Structured Prompt for Reasoning**  Using the template in Figure 9, the module converts ranked candidates into a concise narrative that guides the LLM:

1. **Golden Positives:** verified past interactions $(u, r, p, t)$ to establish behavioral context;
2. **Potential Future Links:** top-ranked, high-confidence candidates;
3. **Ambiguous Links:** mid-ranked, explicitly uncertain candidates to avoid over-reliance on weak signals;
4. **Unlikely Links:** lowest-ranked candidates as negatives.

This structure grounds the LLM's reasoning while preserving the relative strengths of retrieved evidence.

**Dynamic Few-shot Learning from Errors**  When the calibrated probability of the correct answer is below a threshold (wrong/uncertain), and the example bank is not full, the module triggers a diagnostic step: it asks the LLM to *explain why* the gold node $v_{\text{true}}$ is correct given the same context, formats the generated rationale into a compact few-shot example, and prepends it to subsequent prompts (Figure 10). This closed loop distills failures into reusable guidance, steadily improving interpretation of similar graph contexts without any parameter updates.

## 3.4 ADAPTIVE PARAMETERS

One-size-fits-all settings are ill-suited to heterogeneous and evolving TKGs, in which the salience of *temporal activity*, *structural topology*, and *semantic affinity* varies across nodes and over time. We therefore generate, for each query $(u, r, ?, t)$, five runtime hyperparameters—recall capacities $(k_{\text{struc}}, k_{\text{sem}}, k_{\text{time}})$ and scoring weights $(\alpha, \beta)$—so the strategy specializes to $u$'s context at time $t$. The adaptation is anchored by precomputed global statistics $\mathbb{G}$ summarizing the graph's state, ensuring query-level adjustments remain calibrated to the overall environment. The computation of each parameter is detailed below.

**Structural recall capacity ($k_{\text{struc}}$).** We adapt the number of structurally retrieved nodes to the query node's degree: higher-degree nodes warrant broader search. We compute a scaling factor from the node's log-degree, $\ell_u = \log(d_u + 1)$, normalized by the global mean $\mu_{\text{logdeg}} \in \mathbb{G}$:

$$k_{\text{struc}} = \left\lfloor k_{\text{struc}}^{(\text{base})} \cdot (1 + \gamma_{\text{struc}} \cdot (\log(d_u + 1) - \mu_{\text{logdeg}})) \right\rfloor.$$

Here, $\gamma_{\text{struc}}$ is a sensitivity factor. The result is clamped within a reasonable range $[k_{\min}, 2 \cdot k_{\text{struc}}^{(\text{base})}]$ to ensure stability.

**Temporal recall capacity ($k_{\text{time}}$).** We adapt the temporal window to recent activity: when events are sparse, we narrow the look-back to emphasize the most recent nodes. Let $\Delta t_k$ be the gap from $t$ to the $k_{\text{time}}^{(\text{base})}$-th most recent event and let $\mu_{\Delta t} \in \mathbb{G}$ denote the global mean inter-event gap. We set

$$k_{\text{time}} = \left\lfloor k_{\text{time}}^{(\text{base})} \cdot \left(1 - \gamma_{\text{time}} \cdot \tanh\left(\frac{\Delta t_k - \mu_{\Delta t}}{\mu_{\Delta t}}\right)\right) \right\rfloor,$$

where $\gamma_{\text{time}}$ is the scaling factor for time recall. The $\tanh$ function ensures a smooth and bounded adjustment.

**Semantic recall capacity ($k_{\text{sem}}$).** We size the semantic pool by the *concentration* of similarities around $u$. Let $\sigma_u$ be the standard deviation of cosine similarities between $u$ and its top-$2k_{\text{sem}}^{(\text{base})}$ nearest semantic neighbors. A large $\sigma_u$ (steep drop-off) favors a *smaller*, more precise pool; a small $\sigma_u$ (flat tail) favors a *larger* pool. We adjust $k_{\text{sem}}$ relative to the global mean $\mu_\sigma \in \mathbb{G}$:

$$k_{\text{sem}} = \left\lfloor k_{\text{sem}}^{(\text{base})} \cdot \left(1 - \gamma_{\text{sem}} \cdot \tanh\left(\frac{\sigma_u - \mu_\sigma}{\mu_\sigma}\right)\right) \right\rfloor.$$

Here, $k_{\text{sem}}^{(\text{base})}$ is the default size; $\sigma_u$ measures local similarity spread; $\mu_\sigma$ provides global calibration; $\gamma_{\text{sem}} \in (0, 1]$ controls adjustment strength.

**Scoring weight $\alpha$.** $\alpha$ balances the influence of structural vs. semantic evidence in the similarity calculation, which is tuned based on the node's structural connectivity. Well-connected nodes (high $\log(d_u + 1)$) provide more reliable structural signals, so $\alpha$ is increased to more strongly weigh structural similarity:

$$\alpha = \text{CLAMP}\left(\alpha^{(\text{base})} + \delta_\alpha \cdot \tanh\left(\frac{\log(d_u + 1) - \mu_{\text{logdeg}}}{\phi_\alpha}\right), 0.05, 0.95\right).$$

Here, $\delta_\alpha$ controls the maximum adjustment range, and $\phi_\alpha$ is a smoothing factor.

**Scoring weight $\beta$.** $\beta$ balances the historical similarity score ($S_{\text{hist}}$) against the self-similarity score ($S_{\text{self}}$), which is adapted according to the maturity of the query node, defined by its number of past interactions ($n_u$). Nodes with extensive history provide more reliable data for the historical partner score, so $\beta$ is increased. For nodes with sparse history, the model should rely more on the self-similarity:

$$\beta = \text{CLAMP}\left(\beta^{(\text{base})} + \delta_\beta \cdot \tanh\left(\frac{n_u - \mu_{\text{hist}}}{\phi_\beta}\right), 0.05, 0.95\right).$$

$\mu_{\text{hist}} \in \mathbb{G}$ is the average number of historical interactions per node, $\delta_\beta$ is the adjustment range, and $\phi_\beta$ is a smoothing factor.

# 4 EXPERIMENTS

## 4.1 EXPERIMENTAL SETUP

**Datasets** We evaluate our proposed framework on two large-scale temporal knowledge graph (TKG) datasets: ICEWS1819 and GDELT (Zhang et al., 2024) (detailed in Appendix C). ICEWS1819 integrates events from the Integrated Crisis Early Warning System between 2018 and 2019, while GDELT is a massive dataset of global events based on news media. These datasets represent complex, real-world dynamics and are widely adopted benchmarks for TKG reasoning. The statistics of the processed datasets are summarized in Table 3.

**Baselines** We compare our method against two categories of strong baselines. For dynamic graph neural networks, We include state-of-the-art models from the DTGB benchmark[1] (Zhang et al., 2024), namely JODIE (Kumar et al., 2019), DyRep (Trivedi et al., 2019), TGAT (Xu et al., 2020), CAWN (Wang et al., 2022), TCL (Wang et al., 2021), GraphMixer (Cong et al., 2023), and DyGFormer (Yu et al., 2023). For training-free LLM-based methods, we also reproduce three inference-only LLM methods: ICL (Lee et al., 2023), GAD (Lei et al., 2025), and AnRe (Tang et al., 2025). We provide the details of the above baselines in Appendix D.

**Tasks & Evaluation Metrics** We conduct experiments on two fundamental tasks within both transductive and inductive settings. In the inductive setting, we focus on nodes that were not seen during the training of parameter-trained models. For the future link prediction task, we report the standard metrics: Area Under the ROC Curve (AUC-ROC) and Average Precision (AP) scores.

**Implementation Details** We implement our framework using PyTorch and the Transformers library. The LLM backbone is Qwen3-8B[2] (Yang et al., 2025), and the semantic text encoder is the all-mpnet-base-v2[3] model from SentenceTransformers. Structural features are generated by Random Projection Module (Lu et al., 2024) (detailed in Appendix E). The implementation details of parameter settings and evaluation methods are described in Appendix F. All experiments are conducted on an NVIDIA A100 Tensor Core GPU with 80GB of VRAM. Following common practice (Zhang et al., 2024), we chronologically split the data into training (70%), validation (15%), and testing (15%) sets. All results are averaged over three runs with different random seeds, reported as *mean ± standard deviation*.

---

[1]https://github.com/zjs123/DTGB

[2]https://huggingface.co/Qwen/Qwen3-8B

[3]https://huggingface.co/sentence-transformers/all-mpnet-base-v2

Table 1: Under the global negative sampling strategy, the performance comparison of various models on the link prediction task. ♣ represents trained dynamic graph models, and ◇ represents LLM-based methods without training. *tr.* means transductive setting, and *in.* means inductive setting. The best performance within each model type is highlighted in **bold**.

| Type | Model | ICEWS1819 | | | | GDELT | | | |
| --- | --- | --- | --- | --- | --- | --- | --- | --- | --- |
| | | *tr.* | | *in.* | | *tr.* | | *in.* | |
| | | AP | AUC-ROC | AP | AUC-ROC | AP | AUC-ROC | AP | AUC-ROC |
| ♣ | JODIE | 97.52 ± 0.37 | 97.41 ± 1.13 | 93.18 ± 0.39 | 92.85 ± 0.65 | 94.66 ± 0.32 | 95.33 ± 0.20 | 90.12 ± 0.22 | 90.87 ± 0.25 |
| | DyRep | 96.76 ± 0.26 | 96.32 ± 0.27 | 90.65 ± 0.62 | 90.30 ± 0.97 | 94.16 ± 0.17 | 94.53 ± 0.18 | 87.65 ± 0.35 | 88.32 ± 0.39 |
| | TGAT | 99.08 ± 0.32 | 99.04 ± 0.39 | 97.23 ± 0.38 | 97.06 ± 0.54 | 95.72 ± 0.29 | 95.95 ± 0.33 | 92.87 ± 0.25 | 93.51 ± 0.28 |
| | CAWN | 98.86 ± 0.25 | 98.57 ± 0.18 | 97.91 ± 0.27 | 97.74 ± 0.39 | 95.82 ± 0.53 | 96.00 ± 0.61 | 93.59 ± 0.19 | 94.23 ± 0.21 |
| | TCL | **99.27 ± 0.12** | **99.23 ± 0.12** | **97.95 ± 0.10** | **97.78 ± 0.12** | 96.01 ± 0.11 | 96.19 ± 0.08 | **93.74 ± 0.07** | **94.38 ± 0.08** |
| | GraphMixer | 98.71 ± 0.34 | 98.63 ± 0.24 | 96.28 ± 0.64 | 96.05 ± 0.89 | 95.23 ± 0.20 | 95.52 ± 0.18 | 92.08 ± 0.38 | 92.74 ± 0.41 |
| | DyGFormer | 99.01 ± 0.18 | 98.88 ± 0.15 | 96.32 ± 0.09 | 96.13 ± 0.10 | **96.53 ± 0.03** | **96.62 ± 0.03** | 92.61 ± 0.05 | 93.20 ± 0.06 |
| ◇ | ICL | 89.60 ± 0.04 | 88.63 ± 0.04 | 88.33 ± 0.06 | 87.26 ± 0.05 | 90.56 ± 0.06 | 90.33 ± 0.07 | 90.02 ± 0.01 | 89.76 ± 0.01 |
| | GAD | 84.35 ± 1.03 | 82.48 ± 1.03 | 83.50 ± 0.02 | 81.59 ± 0.02 | 85.56 ± 0.64 | 84.71 ± 0.74 | 84.94 ± 2.25 | 84.12 ± 2.40 |
| | AnRe | 90.98 ± 0.09 | 90.00 ± 0.08 | 90.47 ± 0.16 | 89.36 ± 0.19 | 83.71 ± 0.02 | 81.86 ± 0.01 | 80.78 ± 0.12 | 78.43 ± 0.05 |
| | **DyCo-LLM** | **99.18 ± 0.03** | **99.15 ± 0.03** | **99.14 ± 0.06** | **99.13 ± 0.05** | **97.69 ± 0.08** | **97.73 ± 0.09** | **98.85 ± 0.04** | **98.83 ± 0.02** |

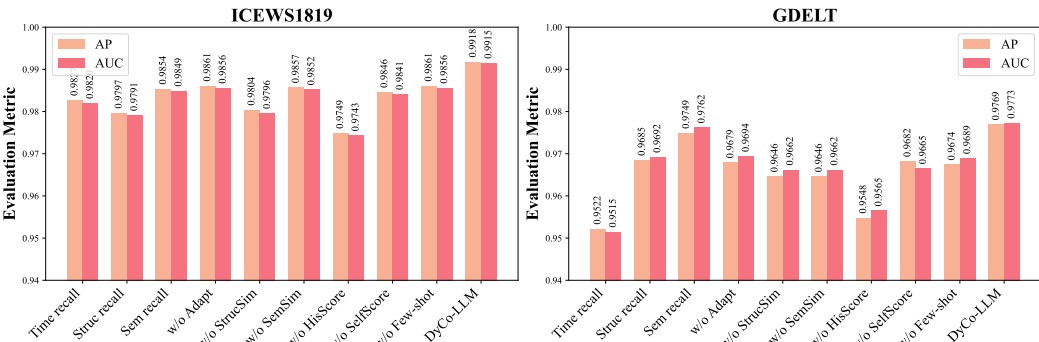

Figure 3: Ablation study of the components in DyCo-LLM.

## 4.2 MAIN RESULTS

Table 1 reports performance on link prediction under *global random* negatives. Dynamic-graph baselines are excerpted from DTGB (Zhang et al., 2024). Experiments show that DyCo-LLM is competitive across configurations. While TCL is slightly better on ICEWS1819 (transductive), DyCo-LLM surpasses all baselines in the remaining settings, with the largest gains in *inductive* setting. Notably, in inductive evaluation DyCo-LLM improves over state of the art by ∼3.3% AP and ∼3.1% ROC-AUC on average. Unlike training-heavy methods, DyCo-LLM is *training-free* and adapts at test time, mitigating overfitting. Its multi-path retrieval plus few-shot self-diagnosis yields query-tailored context and stable generalization. The strong margins suggest that *adaptive inference* aligns better with the real-time dynamics of evolving TKGs.

## 4.3 ABLATION STUDY

We ablate DyCo-LLM on ICEWS1819 and GDELT to isolate the effect of each component (Fig. 3).

**Multi-path recall.** Utilizing any single path (*time-active*, *structural*, or *semantic*) substantially degrades performance, showing that recency, topology, and content provide complementary evidence. **Adaptive retrieval.** Fixing recall budgets (**w/o Adapt**) yields further drops, confirming the need for query-time adjustment to track temporal shift. **Balanced scoring.** Forcing $\alpha = 0$ (structural-only) or $\alpha = 1$ (semantic-only) consistently hurts accuracy, indicating that fusing both views is necessary. Similarly, $\beta = 0$ (no history) or $\beta = 1$ (no self-similarity) reduces performance, verifying the complementarity of historical aggregation and direct affinity. **Reflective few-shot.** Removing dynamic few-shot learning (**w/o few-shot**) causes notable declines across datasets, evidencing the gains from turning errors into corrective exemplars.

Table 2: Under the recall-pool negative sampling strategy, the performance comparison of LLM-based models on the link prediction task. *tr.* means transductive setting, and *in.* means inductive setting. The best performance is highlighted in **bold**.

| Model | ICEWS1819 | | | | GDELT | | | |
| --- | --- | --- | --- | --- | --- | --- | --- | --- |
| | *tr.* | | *in.* | | *tr.* | | *in.* | |
| | AP | AUC-ROC | AP | AUC-ROC | AP | AUC-ROC | AP | AUC-ROC |
| ICL | 86.48 ± 0.12 | 85.51 ± 0.07 | 85.32 ± 0.09 | 84.87 ± 0.06 | 74.62 ± 0.07 | 74.07 ± 0.06 | 74.58 ± 0.08 | 74.02 ± 0.05 |
| GAD | 83.53 ± 0.72 | 82.35 ± 0.81 | 83.41 ± 0.65 | 82.28 ± 0.75 | 71.45 ± 0.16 | 70.30 ± 0.11 | 71.38 ± 0.14 | 70.25 ± 0.09 |
| AnRe | 91.47 ± 0.34 | 91.26 ± 0.31 | 91.39 ± 0.28 | 91.19 ± 0.28 | 80.24 ± 0.02 | 79.90 ± 0.21 | 81.19 ± 0.04 | 80.15 ± 0.18 |
| **DyCo-LLM** | **94.19 ± 0.06** | **94.10 ± 0.05** | **94.14 ± 0.05** | **94.06 ± 0.04** | **87.24 ± 0.08** | **87.12 ± 0.04** | **87.97 ± 0.07** | **87.88 ± 0.03** |

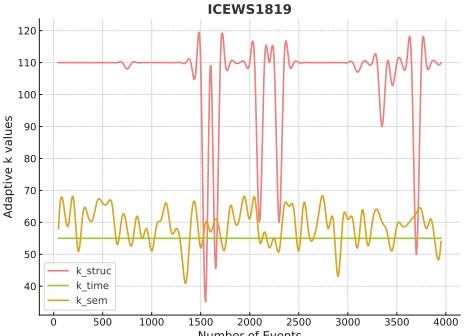 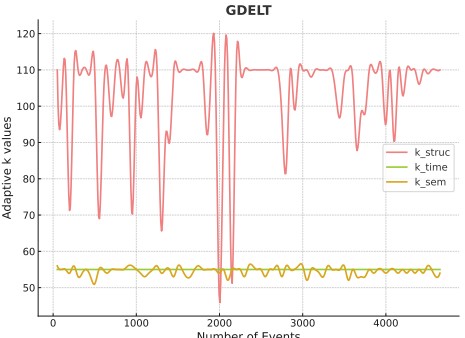

Figure 4: Under ICEWS1819 and GDELT, the adaptive changes of three recall quantities k as the test progresses.

Overall, DyCo-LLM's *multi-path recall*, *adaptive retrieval*, *balanced scoring*, and *self-diagnosis reasoning* are all indispensable for effective performance on streaming temporal knowledge graphs, enabling it to excel in dynamic environments with high predictive accuracy.

### 4.4 ANALYSIS AND DISCUSSION

**Impact of Negative Sampling Strategies** Based on the recall pool obtained from the multi-path recall module, we propose a more challenging task formulation, which evaluates using Recall-pool Negative Sampling (see Appendix F for details). Switching from global random negatives to recall-pool negatives causes a sharp drop in AC/AUC for all methods (see Table 2), because global sampling produces many trivially wrong distractors that LLMs can reject by surface cues, whereas recall-pool sampling draws hard negatives from the candidate set retrieved around the query—semantically/structurally plausible and thus much harder to disambiguate. Despite this harder regime, DyCo-LLM remains the strongest LLM-based model on both datasets and settings, and—crucially—its advantage persists in the inductive (*in.*) setting, indicating stable generalization.

**Interpretability of adaptive parameters** In Figure 4, we show the recall curves for three strategies as the number of recalls $k$ changes. Figure 8 illustrates the frequency distribution of alpha and beta values. Across the ICEWS1819 and GDELT datasets, the structural budget $k_{\text{struc}}$ exhibits the most variation, while $k_{\text{time}}$ remains stable and $k_{\text{sem}}$ fluctuates slightly. This indicates that structural factors, such as degree bursts and community rewiring, are the main sources of non-stationarity, as event rates are stable and semantic embeddings are well-calibrated. To accommodate these changes, the engine expands $k_{\text{struc}}$ for structural shifts while maintaining a consistent temporal window and fine-tuning semantic recall as necessary. The heatmaps for parameters $(\alpha, \beta)$ peak around $\alpha \approx 0.73$–$0.75$ and $\beta \approx 0.65$, suggesting a preference for structural similarity and history aggregation. By emphasizing structural factors, we can eliminate semantically plausible but structurally implausible distractors. A moderately high $\beta$ enhances decision stability by aggregating reliable historical partners without losing the self-term for emerging nodes. Overall, the adaptations align with the data, reflecting structure-dominated dynamics alongside steady activity and semantics.

## 5 RELATED WORK

**LLM for temporal knowledge graphs.** Harnessing their deep semantic and reasoning capacities, large language models (LLMs) have opened a new paradigm for temporal knowledge-graph (TKG) forecasting. Early attempts (Peters et al., 2019; Han et al., 2023; Yang et al., 2024; Xu et al., 2024) repurposed pre-trained LMs as "temporal encoders": historical quadruples were linearized into token sequences and fed to masked-language-modeling objectives to distill event embeddings, empirically confirming the utility of textual priors for temporal inference. Follow-ups (Jiang et al., 2023; Tan et al., 2023; Yuan et al., 2023) further unified temporal and structural signals by devising time-aware positional encodings and relation-specific prompt templates, enabling LLMs to discriminate causal chronologies even in zero-shot regimes. Among them, zrLLM (Ding et al., 2024) and LLM-DA (Ye et al., 2024) introduced plug-and-play "temporal adapters" for domain adaptation, yet their reliance on full-parameter fine-tuning incurs prohibitive computational overhead. Recent research has pivoted toward lightweight prompting: Shi et al. (2023) and Zhang et al. (2025) recast link prediction as autoregressive generation, exploiting ICL (Lee et al., 2023) to compress training costs; GenTKG (Liao et al., 2024) activates cross-domain generalization via few-shot instruction tuning; ONSEP (Yu et al., 2024) co-evolves an LLM with the TKG update stream for dynamic environment adaptation; CoH (Xia et al., 2024) incorporates a higher-order history module that compresses multi-hop paths into reusable semantic fragments, further elevating the expressive ceiling of graph models. Despite these advances, the spotlight remains on quadruple-level event prediction, leaving the DyTAG scenario—where edges and nodes carry rich textual descriptions—largely untouched. Moreover, prevailing approaches adopt day-level or snapshot granularities, rendering them insufficient to portray millisecond-grade interaction dynamics.

**Dynamic graph neural networks.** Confined to pure structural–temporal signals, dynamic graph learning has crystallized into two dominant technical routes. The first, "snapshot-based" methods (Pareja et al., 2019; Sankar et al., 2019) , segment the evolving network into equally spaced snapshots, apply standard GNN aggregation, and subsequently model temporal dependencies with RNNs or Transformers. DySAT (Sankar et al., 2019) simultaneously updates structural and temporal contexts via self-attention, attaining state-of-the-art dynamic link-prediction accuracy, yet is shackled by snapshot granularity and memory explosion. The second, "continuous-time" paradigm (Kumar et al., 2019; Trivedi et al., 2018), employs temporal point processes to parameterize edge-generation intensities; TGAT (Xu et al., 2020) and TGN (Rossi et al., 2020) map continuous timestamps into embedding space through time encoders and couple them with memory modules, achieving constant-time complexity and superior scalability. To mitigate label scarcity, DyGLib (Yu et al., 2023) recently unified 13 dynamic-graph datasets under robust evaluation protocols, fostering standardization. The approaches mentioned above face significant limitations: they cannot adapt to new data during inference due to their fixed parameters, and they are ineffective at integrating textual dynamics with structural evolution. As a result, their performance degrades over time as the knowledge graph expands and its distribution shifts. In contrast, our method overcomes this limitation via an online-coupled pipeline of "dynamic memory parameter → adaptive memory retrieval strategies → self-diagnosis reasoning," which injects large-model semantics and continuous-time signals into the same outer-loop optimization without requiring any training process.

## 6 CONCLUSION

This paper introduce a training-free Dynamic Cognition paradigm, DyCo-LLM, for temporal knowledge graphs. DyCo-LLM performs in-situ test-time adaptation: an adaptive diagnoser sets runtime hyperparameters from local context, multi-path recall gathers temporal, structural, and semantic evidence, dynamic fusion balances history aggregation with self-similarity, and self-diagnosis converts errors into compact, reusable exemplars. Without updating weights, the model tailors reasoning to the evolving graph and keeps prompts concise yet informative. Experiments deliver state-of-the-art results across benchmarks, with especially strong gains in the inductive regime—where prior methods tend to overfit historical neighborhoods—underscoring the importance of live, in-situ adaptation over static training or fixed . Future work will extend DyCo-LLM to support more temporal KGs and dynamic (text-attributed) graph datasets, and refine the coupling between diagnostics and candidate generation.

# 7 ETHICS STATEMENT

All experiments utilized publicly available or synthetically generated datasets, which underwent anonymization procedures. We thoroughly assessed potential model biases and societal implications, while also evaluating the safety of generated content. The authors declare no conflicts of interest. The code and data associated with this study have been open-sourced to enhance transparency and reproducibility.

# 8 REPRODICIBILITY STATEMENT

We have open-sourced all experimental code and data in an anonymous repository. All experimental results can be reproduced using the provided code.

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

Table 3: Statistics of the datasets.

| Dataset | # Node | # Edge | Edge Categories | # Timestamp | Domain | Text Attributes |
|---------|--------|--------|-----------------|-------------|--------|-----------------|
| ICEWS1819 | 31,796 | 1,100,071 | 266 | 730 | Knowledge graph | Node & Edge |
| GDELT | 6,786 | 1,339,245 | 237 | 2,591 | Knowledge graph | Node & Edge |

## A  ALGORITHM

---

**Algorithm 1** Dynamic Score Calculation

---

**Require:** Query node $u$ and timestamp $t$
**Require:** Candidate set $C$
**Require:** Node history $\mathcal{H}$
**Require:** Semantic embeddings $\mathbf{E}_{\text{sem}}$
**Require:** Structural embeddings $\mathbf{E}_{\text{struc}}$
**Require:** Time decay rate $\lambda_{\text{time}}$
**Require:** Adaptive weights $\alpha, \beta$
**Ensure:** Final scores $S_{\text{final}}$ for all $v \in C$
1: $\mathcal{P} \leftarrow \text{GetHistoricalPartners}(u, t, \mathcal{H})$
2: $\mathbf{W} \leftarrow \emptyset$
3: **for** $p \in \mathcal{P}$ **do**
4:     $f_{up} \leftarrow \text{CountInteractions}(u, p, t, \mathcal{H})$
5:     $t_{\text{last}} \leftarrow \text{GetLastInteractionTime}(u, p, t, \mathcal{H})$
6:     $\Delta t \leftarrow t - t_{\text{last}}$
7:     $w_p \leftarrow f_{up} \cdot \exp(-\lambda_{\text{time}} \cdot \Delta t)$
8:     $\mathbf{W} \leftarrow \mathbf{W} \cup w_p$
9: **end for**
10: $\mathbf{W} \leftarrow \text{Normalize}(\mathbf{W})$
11: **for** $v \in C$ **do**
12:     $S_{\text{hist}} \leftarrow 0$
13:     **for** $j = 0$ to $|\mathcal{P}| - 1$ **do**
14:         $p \leftarrow \mathcal{P}[j]$
15:         $s_{\text{sem}} \leftarrow \text{CosineSimilarity}(\mathbf{E}_{\text{sem}}[v], \mathbf{E}_{\text{sem}}[p])$
16:         $s_{\text{struc}} \leftarrow \text{CosineSimilarity}(\mathbf{E}_{\text{struc}}[v], \mathbf{E}_{\text{struc}}[p])$
17:         $s_{\text{total}} \leftarrow \alpha \cdot s_{\text{struc}} + (1 - \alpha) \cdot s_{\text{sem}}$
18:         $S_{\text{hist}} \leftarrow S_{\text{hist}} + s_{\text{total}} \cdot \mathbf{W}[j]$
19:     **end for**
20:     $s_{\text{sem}}^{(self)} \leftarrow \text{CosineSimilarity}(\mathbf{E}_{\text{sem}}[v], \mathbf{E}_{\text{sem}}[u])$
21:     $s_{\text{struc}}^{(self)} \leftarrow \text{CosineSimilarity}(\mathbf{E}_{\text{struc}}[v], \mathbf{E}_{\text{struc}}[u])$
22:     $S_{\text{self}} \leftarrow \alpha \cdot s_{\text{struc}}^{(self)} + (1 - \alpha) \cdot s_{\text{sem}}^{(self)}$
23:     $S_{\text{final}}[v] \leftarrow \beta \cdot S_{\text{hist}} + (1 - \beta) \cdot S_{\text{self}}$
24: **end for**
        **return** $S_{\text{final}}$

---

## B  THEORETICAL ANALYSIS

We formalize when and why **DyCo-LLM** improves test-time link prediction on temporal knowledge graphs (TKGs). Let $G_{<t}$ denote the history up to time $t$. For a query $(u, r, ?, t)$, let the (unknown) Bayes-optimal tail be $v^\star \in \arg\max_{v \in \mathcal{E}} \Pr\big((u, r, v, t) \in G \,\big|\, G_{<t}\big)$. Our dynamic context engine constructs a candidate set $C \subseteq \mathcal{E}$ via multi-path recall and computes a score $S_{\text{final}}(v) = \beta S_{\text{hist}}(v) + (1-\beta) S_{\text{self}}(v)$ with $S_{\text{hist}}(v) = \sum_{p \in P(u)} \Big[\alpha \sin_{\text{str}}(v, p) + (1-\alpha) \sin_{\text{sem}}(v, p)\Big] W(p)$ and $S_{\text{self}}(v) = \alpha \sin_{\text{str}}(v, u) + (1 - \alpha) \sin_{\text{sem}}(v, u)$. Weights $W(p) \propto \text{Count}(u, p) \exp\{-\lambda_{\text{time}}(t - t_{\text{last}}(u, p))\}$ emphasize frequent and recent partners.

**Assumptions.** We adopt mild regularity conditions commonly used in nonparametric estimation on dynamic graphs:

(A1) **Recall coverage.** For any query, with probability at least $1 - \delta$, the true tail is recalled: $\Pr\left(v^\star \in C \mid G_{<t}\right) \geq 1 - \delta$.

(A2) **Score–probability monotonicity (local).** There exist $g_{\text{hist}}, g_{\text{self}}$ such that $\mathbb{E}[S_{\text{hist}}(v) \mid G_{<t}] = g_{\text{hist}}\left(\Pr(v \mid G_{<t})\right)$, $\mathbb{E}[S_{\text{self}}(v) \mid G_{<t}] = g_{\text{self}}\left(\Pr(v \mid G_{<t})\right)$, and both $g_.$ are strictly increasing on the support of $\Pr(v \mid G_{<t})$.

(A3) **Bounded noise.** The zero-mean deviations $\varepsilon_{\text{hist}}(v) = S_{\text{hist}}(v) - \mathbb{E}[S_{\text{hist}}(v) \mid G_{<t}]$, $\varepsilon_{\text{self}}(v) = S_{\text{self}}(v) - \mathbb{E}[S_{\text{self}}(v) \mid G_{<t}]$ are sub-Gaussian with proxy variance at most $\sigma^2$, independent across $v$ conditional on $G_{<t}$.

**Proposition 1** (Top-1 consistency under coverage and monotonicity). *Under (A1)–(A3), let $v_{(1)} = \arg\max_{v \in C} S_{\text{final}}(v)$. Then*

$$\Pr\left(v_{(1)} = v^\star\right) \geq (1 - \delta) \cdot \left(1 - 2 \exp\left\{-\frac{\Delta^2}{8\sigma^2}\right\}\right),$$

*where* $\Delta = \min_{v \in C \setminus \{v^\star\}}\left(\mu_{\text{final}}(v^\star) - \mu_{\text{final}}(v)\right)$ *and* $\mu_{\text{final}}(v) = \beta \mathbb{E}[S_{\text{hist}}(v) \mid G_{<t}] + (1 - \beta) \mathbb{E}[S_{\text{self}}(v) \mid G_{<t}]$.

*Sketch.* On the event $v^\star \in C$, sub-Gaussian concentration and a union bound imply that the empirical score ranking matches the mean-score ranking with probability at least $1 - 2\exp\{-\Delta^2/(8\sigma^2)\}$. Since mean scores are strictly increasing transforms of the true tail probability by (A2), the Bayes-optimal element attains the largest mean score. Multiplying by $(1 - \delta)$ from (A1) yields the claim. $\square$

**Remark 1** (Why in-situ adaptation helps). *The adaptive budgets $k_{\text{struc}}, k_{\text{sem}}, k_{\text{time}}$ directly control the coverage term $1 - \delta$: in sparse/quiet temporal regimes, increasing $k_{\text{time}}$ (or shifting weight to structural/semantic paths) raises the chance that $v^\star$ is recalled; in dense/active regimes, smaller budgets reduce distractors and shrink $\sigma^2$. Similarly, adjusting $\alpha, \beta$ modifies the signal gap $\Delta$ by emphasizing the more reliable channel (history vs. self) for the current node maturity and degree. Both effects tighten the bound in Proposition 1.*

**Proposition 2** (Connection to kernel intensity estimation). *If events follow a self-exciting process where the conditional link intensity admits a separable form $\lambda_{u \rightarrow v}(t) \propto \sum_{p \in P(u)} \kappa_\tau(t - t_{up}) \phi_{str}(v, p)^\alpha \phi_{sem}(v, p)^{1-\alpha}$, then $S_{\text{hist}}(v)$ is a Monte-Carlo estimator of $\lambda_{u \rightarrow v}(t)$ with kernel $\kappa_\tau(\Delta t) = \exp(-\lambda_{\text{time}}\Delta t)$. Consequently, ranking by $S_{\text{hist}}$ is Fisher-consistent for ranking by $\lambda$.*

*Sketch.* Replace $\text{Count}(u, p)$ by discrete sampling from past partners and interpret the exponential time decay as a positive kernel. The similarity fusion provides a plug-in estimator of the mark-dependent similarity $\phi$. Up to a normalization constant, $S_{\text{hist}}$ approximates $\lambda$. $\square$

**Proposition 3** (Bias reduction via position calibration). *Let $Z \in \{A, B\}$ denote the displayed position of $v^\star$ in a binary prompt. Suppose the LLM's reported probability obeys $\Pr(\text{choose } A \mid Z{=}A) = \pi + b$ and $\Pr(\text{choose } B \mid Z{=}B) = \pi - b$, where $\pi$ is the true preference for $v^\star$ and $b$ is an additive position bias. The calibrated estimator $\hat{\pi} = \frac{1}{2}\left(P_1(A) + P_2(B)\right)$ is unbiased: $\mathbb{E}[\hat{\pi}] = \pi$, and has variance $\text{Var}(\hat{\pi}) = \frac{1}{2}\text{Var}(P_1) - \frac{1}{4}\left(\mathbb{E}[P_1] - \mathbb{E}[P_2]\right)^2$.*

*Sketch.* Direct calculation shows the opposite-sign bias terms cancel after the role-swap; the variance follows from independence of the two passes conditioned on the context. $\square$

**Takeaway.** Under mild coverage/monotonicity/noise conditions, DyCo-LLM's adaptive recall and scoring tighten a PAC-style success bound by increasing the signal gap and lowering noise, while the calibration and reflective few-shot reduce systematic bias and improve effective signal-to-noise over time.

## C    DETAILS OF DATASETS

**GDELT**[4] is constructed from the Global Database of Events, Language, and Tone (GDELT) project, which monitors political events and activities across the world in near real-time. Nodes corre-

---

[4]https://www.gdeltproject.org/

spond to political actors (e.g., United States, Kim Jong Un) and are represented by their names. Edges denote types of interaction or relationship between these actors (e.g., MAKE_STATEMENT, ENGAGE_IN_DIPLOMACY). The textual attributes of edges are derived from the verbal descriptions of these relation types. Each event is timestamped with 15-minute granularity, resulting in a high-resolution temporal graph that captures rapidly evolving political dynamics.

**ICEWS1819**[5] is built from the Integrated Crisis Early Warning System (ICEWS), covering events from January 1, 2018, to December 31, 2019. Nodes represent political entities and are annotated with composite textual features including name, sector, and nationality. Relations capture discrete political or military actions, with edge text formed from the semantic description of each event type. Events are ordered at a daily granularity. Key distinctions from GDELT include: (1) a coarser temporal resolution (24-hour intervals), and (2) a significantly larger node set (approximately 4× that of GDELT), reflecting a sparser and more diversified interaction network.

# D DETAILS OF BASELINES

## D.1 TEMPORAL GRAPH MODELS

**JODIE (Kumar et al., 2019)** employs two interconnected recurrent networks to model the temporal evolution of entity representations. It incorporates a projection mechanism that forecasts the future trajectory of node embeddings, enabling predictions about both entity states and their potential future interactions.

**DyRep (Trivedi et al., 2019)** introduces a recurrent architecture that updates node representations following each interaction event. It implements a temporal point process framework to capture underlying dynamics of network evolution, enhanced by a temporal-attentive mechanism that encodes time-varying structural patterns into node embeddings to drive the dynamic evolution of the graph.

**TGAT (Xu et al., 2020)** Utilizing self-attention as its core component, this method effectively aggregates temporal-topological neighborhood features while modeling complex time-feature interactions. It incorporates a functional time encoding technique grounded in Bochner's theorem from harmonic analysis to capture rich temporal patterns in dynamic graphs.

**CAWN (Wang et al., 2022)** employs an anonymization strategy through sampled temporal walks to investigate causal relationships in network dynamics and generate inductive node identifiers. These sampled walks are subsequently encoded and aggregated through neural networks to produce final node representations.

**TCL (Wang et al., 2021)** is a dual-stream encoder architecture that processes temporal neighborhoods of interacting nodes separately. It introduces a graph-topology-aware Transformer that integrates both structural topology and temporal information, supplemented with cross-attention mechanisms to capture relevance between target nodes.

**GraphMixer (Cong et al., 2023)** Demonstrating the efficacy of fixed-time encoding functions, this architecture comprises three key components: a link encoder that summarizes temporal interaction information, a node encoder that aggregates node features, and a link prediction classifier that operates on the encoded representations.

**DyGFormer (Yu et al., 2023)** learns node representations from historical first-hop interactions using a neighbor co-occurrence encoding scheme that captures correlations between nodes based on their interaction sequences. It introduces a patching technique that segments long sequences into manageable patches, enabling effective utilization of extended historical context.

## D.2 LLM-BASED METHODS

All LLM-based methods in our experiments employed Qwen3-8B as the base model. Qwen3 is a new generation of open-source large language model series by Alibaba, launched in April 2025. Its core

---

[5]https://dataverse.harvard.edu/dataverse/icews

innovation lies in the adoption of a 'Mixture of Experts' (MoE) architecture and a 'Mixture Inference' mode, which can dynamically switch between deep reasoning in 'thinking mode' and quick response in 'non-thinking mode' based on task complexity, thereby balancing high performance and efficiency.

**ICL (Lee et al., 2023)** formulates temporal knowledge graph forecasting as an in-context learning problem for LLMs, entirely avoiding task-specific training. It operates through a three-stage pipeline: retrieving relevant historical facts conditioned on the query, converting these facts and the query into a structured textual prompt, and decoding the LLM's output token probabilities into a ranked list of candidate entities.

**GAD (Lei et al., 2025)** proposes a multi-agent system that leverages collaborative LLMs to address predictive tasks on dynamic text-attributed graphs. It incorporates global and local summary agents to generate domain-specific and node-specific knowledge, enhancing transferability across diverse domains. A knowledge reflection mechanism is also introduced to enable adaptive updates, maintaining a unified and self-consistent architecture without dataset-specific training.

**AnRe (Tang et al., 2025)** proposes a training-free reasoning method for temporal knowledge graph forecasting that leverages semantic-driven clustering and dual history extraction. It retrieves similar historical events through entity clustering and constructs comprehensive contexts by integrating both long-term and short-term event chains. By using large language models to generate analogical reasoning examples from these contexts, AnRe enables few-shot learning of historical patterns for accurate prediction of future events without dataset-specific training.

## E    RANDOM PROJECTION

The Random Projection Module is responsible for generating dynamic structural node embeddings without training (Lu et al., 2024). It maintains a set of trainable parameters $\mathbf{P}^{(l)} \in \mathbb{R}^{N \times d}$ for each layer $l$ (where $N$ is the number of nodes and $d$ is the projection dimension). The module is updated incrementally with each new batch of events $(u_i, v_i, t_i)$.

For a new interaction $(u, v, t)$, the update for the higher-order projections $(l \geq 1)$ applies a time-decayed message passing step:

$$\mathbf{P}^{(l)}[u] \leftarrow \mathbf{P}^{(l)}[u] + \exp(-\lambda \cdot (t - t_{\text{prev}})) \cdot \mathbf{P}^{(l-1)}[v]$$

$$\mathbf{P}^{(l)}[v] \leftarrow \mathbf{P}^{(l)}[v] + \exp(-\lambda \cdot (t - t_{\text{prev}})) \cdot \mathbf{P}^{(l-1)}[u]$$

where $\lambda$ is a time decay factor and $t_{\text{prev}}$ is the time of the last update. The final structural embedding for a node is the concatenation of its projections across all layers: $\mathbf{e}_{\text{struc}} = \|_{l=0}^{L} \mathbf{P}^{(l)}[u]$. In the experimental setup, the structural vector dimension factor is set to 10, the maximum hop count for random walks is set to 3, and the time decay weight is $1e - 7$.

## F    ADDITIONAL IMPLEMENTATION DETAILS

**Parameter Setting** To ensure the lower limit of the adaptive capacity of our dynamic parameter module, we determine a baseline value for each parameter through preliminary experiments and experience as the starting point for changes. In the experimental setup, the baseline quantities for recall in three dimensions are 55, and the baseline value of $\alpha, \beta$ is 0.5. Additionally, the time decay rate for historical partner weights is set to 0.01. The future positive sample ratio, based on sample scores, is 0.15, and the negative sample ratio is 0.5. To balance the length of context and the completeness of historical interaction information, we set the number of golden positive samples to 100 and provide 3-shot examples in the few-shot construction module as context input for the LLM.

**Negative Sampling Strategy** For the final prediction, the LLM is presented with a binary choice between the ground-truth node $v_{\text{true}}$ and a carefully sampled negative node $v_{\text{neg}}$. To ensure a challenging and informative comparison, we employ different negative sampling strategies tailored to the task: (1) **Global Sampling:** $v_{\text{neg}}$ is sampled randomly from all nodes in the graph, excluding $u$ and $v_{\text{true}}$; (2) **Recall-pool Sampling (Proposed):** $v_{\text{neg}}$ is sampled from the full recall candidate set $C$,

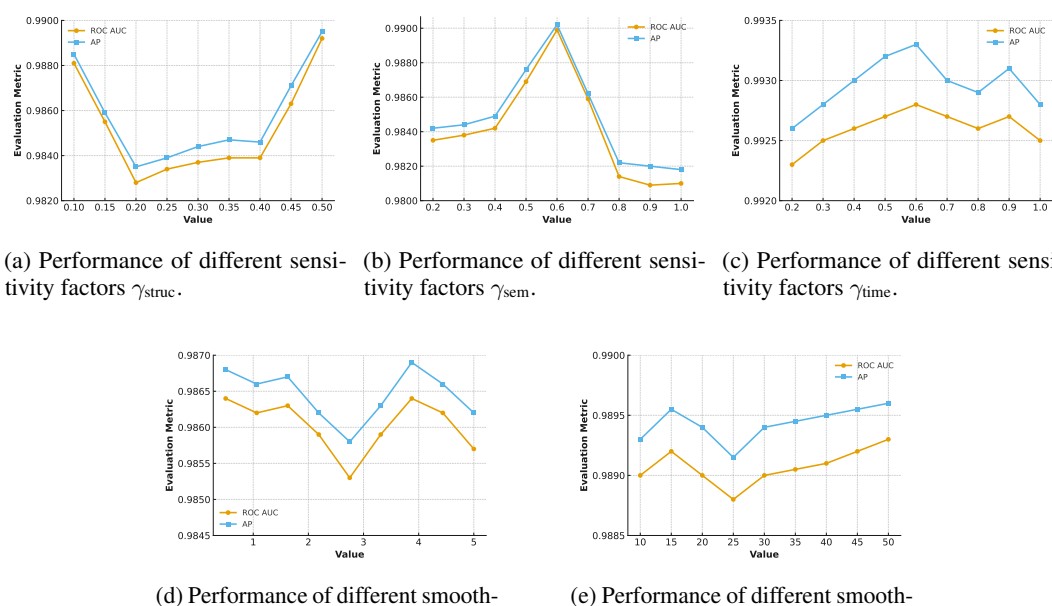

(a) Performance of different sensitivity factors $\gamma_{\text{struc}}$.

(b) Performance of different sensitivity factors $\gamma_{\text{sem}}$.

(c) Performance of different sensitivity factors $\gamma_{\text{time}}$.

(d) Performance of different smoothing factors $\phi_\alpha$.

(e) Performance of different smoothing factors $\phi_\beta$.

Figure 5: Hyperparameter sensitivity analysis under ICEWS1819.

which contains nodes deemed relevant by at least one recall strategy. This ensures $v_{\text{neg}}$ is a plausible but incorrect alternative, making the LLM's choice more discriminative and educationally valuable for the few-shot learner.

**Position Bias Calibration**   Instead of relying solely on the model's generated text output, which can be brittle and inconsistent, we precisely measure its confidence by analyzing the probability distribution over the choice tokens. For each binary choice presented to the LLM, we obtain the logits for the tokens 'A' and 'B' from the final output position. We then apply a softmax function to these logits to obtain a normalized probability distribution, representing the model's confidence in each option. To mitigate positional bias, we perform two inference passes: (1) Pass 1: Present the choice as 'A: $v_{\text{true}}$, B: $v_{\text{neg}}$'; (2) Pass 2: Present the choice as 'A: $v_{\text{neg}}$, B: $v_{\text{true}}$'.

Let $P_1(A)$ be the probability of choosing A from the first pass (i.e., the probability for $v_{\text{true}}$), and $P_2(B)$ be the probability of choosing B from the second pass (i.e., also the probability for $v_{\text{true}}$). The final calibrated probability for $v_{\text{true}}$ is calculated as:

$$P_{\text{final}}(v_{\text{true}}) = \frac{P_1(A) + P_2(B)}{2}.$$

This calibrated score provides a more robust and unbiased estimate of the model's confidence in the true link.

## G   HYPERPARAMETER SENSITIVITY ANALYSIS

As shown in Figure 5, we investigate how the five sensitivity/smoothing factors affect DyCo-LLM's adaptive recall and scoring. Recall that the final score for a candidate $v$ is $S_{\text{final}}(v) = \beta\, S_{\text{hist}}(v) + (1 - \beta)\, S_{\text{self}}(v)$, where $S_{\text{hist}}$ aggregates structural/semantic similarities to the query's historical partners with an exponential time decay, and $S_{\text{self}}$ measures the direct structural/semantic similarity to the query node. The three adaptive recall budgets $k_{\text{struc}}, k_{\text{sem}}, k_{\text{time}}$ are modulated by node degree and local inter-event sparsity. We report AP/AUC as each factor varies while fixing others at their default.

**Impact of $\gamma_{\text{struc}}$.**   $\gamma_{\text{struc}}$ controls how aggressively the structural recall budget $k_{\text{struc}}$ scales with the (centered) log-degree of the query node. When $\gamma_{\text{struc}}$ is very small, $k_{\text{struc}}$ is nearly fixed; the recall for high-degree nodes is slightly *under*-provisioned, while low-degree nodes may be *over*-provisioned

with weak structural neighbors. As $\gamma_{\text{struc}}$ increases from this regime, the first effect dominates and AP/AUC *decrease*: we reduce recall on some low-degree queries (removing a few helpful neighbors) before we have granted enough budget to high-degree cases. After this transient, the curve becomes flat because the allocation approaches a balance between high- and low-degree nodes. Pushing $\gamma_{\text{struc}}$ further finally benefits high-degree queries—where structural evidence is abundant—so the metrics rise again. In short, the trajectory "drop $\rightarrow$ plateau $\rightarrow$ rise" reflects (i) early misallocation for low-degree nodes, (ii) a compensated middle zone, and (iii) late-stage gains on high-degree queries where structural recall is most informative.

**Impact of $\gamma_{\text{sem}}$.** $\gamma_{\text{sem}}$ adjusts the semantic recall budget $k_{\text{sem}}$ with degree-sensitive scaling. If $\gamma_{\text{sem}}$ is too small, semantic recall is conservative and fails to surface enough conceptually related but structurally distant candidates—hurting cold/low-degree queries where semantics is the primary signal in both $S_{\text{hist}}$ and $S_{\text{self}}$. If it is too large, we recall many semantically similar but structurally irrelevant distractors, which inflates the candidate pool entropy and weakens the signal gap $\Delta$ between the correct answer and its competitors in Proposition 1, degrading AP/AUC. A mid-range $\gamma_{\text{sem}}$ trades off coverage and noise, giving the best scores.

**Impact of $\gamma_{\text{time}}$.** $\gamma_{\text{time}}$ governs how $k_{\text{time}}$ reacts to local inter-event sparsity via a squashed (e.g., $\tanh$) mapping. At small values, recent-activity recall underfits burstiness and misses fresh targets; at moderate values, $k_{\text{time}}$ expands during bursts and contracts in quiet periods, maximizing the chance that the true tail $v^\star$ is covered while avoiding excessive distractors—hence a peak in AP/AUC. When $\gamma_{\text{time}}$ grows further, the system becomes over-reactive to short-term fluctuations: the candidate set oscillates across timestamps, and the induced variance in $S_{\text{hist}}$ (through changing partner overlaps) yields metric fluctuations and a slight decline beyond the optimum.

**Impact of $\phi_\alpha$.** $\phi_\alpha$ controls how the mixing weight $\alpha$ (structure vs. semantics inside both $S_{\text{hist}}$ and $S_{\text{self}}$) is smoothed as a function of degree. At very small $\phi_\alpha$, $\alpha$ is close to a global constant; beginning to increase $\phi_\alpha$ nudges $\alpha$ away from its well-tuned baseline and *initially* reduces AP/AUC. As $\phi_\alpha$ continues to grow, $\alpha$ better adapts to node regimes (more structural for high-degree, more semantic for low-degree), which restores and improves accuracy. If $\phi_\alpha$ is pushed too high, the adaptation over-amplifies regime differences and becomes brittle to noise in degree statistics, causing another drop. This explains the "decrease $\rightarrow$ increase $\rightarrow$ decrease" pattern.

**Impact of $\phi_\beta$.** $\phi_\beta$ regulates how the fusion weight $\beta$ (history vs. self) depends on node maturity (the amount and recency of past interactions). When $\phi_\beta$ starts increasing from a small value, the estimator relies more aggressively on imperfect maturity signals; with few past events, maturity is noisy and the learned $\beta$ fluctuates, leading to oscillatory performance. As $\phi_\beta$ becomes larger, the smoothing stabilizes $\beta$ and consistently privileges $S_{\text{hist}}$ for truly mature nodes while keeping sufficient $S_{\text{self}}$ for immature ones; the adaptation hence delivers a steady improvement in AP/AUC in the later range.

## H  CASE STUDY: SELF-DIAGNOSIS REASONING IN DYCO-LLM

In Figure 6, we analyze the *self-diagnosis reasoning* module on concrete test cases from our case-study figure, focusing on how the module corrects errors *without* any entity semantics (all entities are numeric IDs).[6] The binary query uses the standardized form "Which node is more likely to link with node $u$? A: $v_{\text{true}}$; B: $v_{\text{neg}}$," and the reflective step constructs a diagnostic rationale when the model answers incorrectly or with low confidence; the generated rationale is stored as a few-shot example for subsequent queries.

**Case 1 (error $\rightarrow$ correction by structure-aware rationale).** For query node $u{=}4910$ (choices 66 vs. 4408), the model first answers "$B$" (i.e., 4408). The self-diagnosis module then produces a concise analysis pointing to repeated historical interactions between 4910 and 66 (via the relation "Accuse" at timestamp 616) and the absence of any evidence for 4408 in the provided context. This rationale is appended to the few-shot bank and helps flip similar future decisions toward the history-consistent choice.

---

[6]At inference time the prompt explicitly presents entities as IDs; no names or textual descriptions are given.

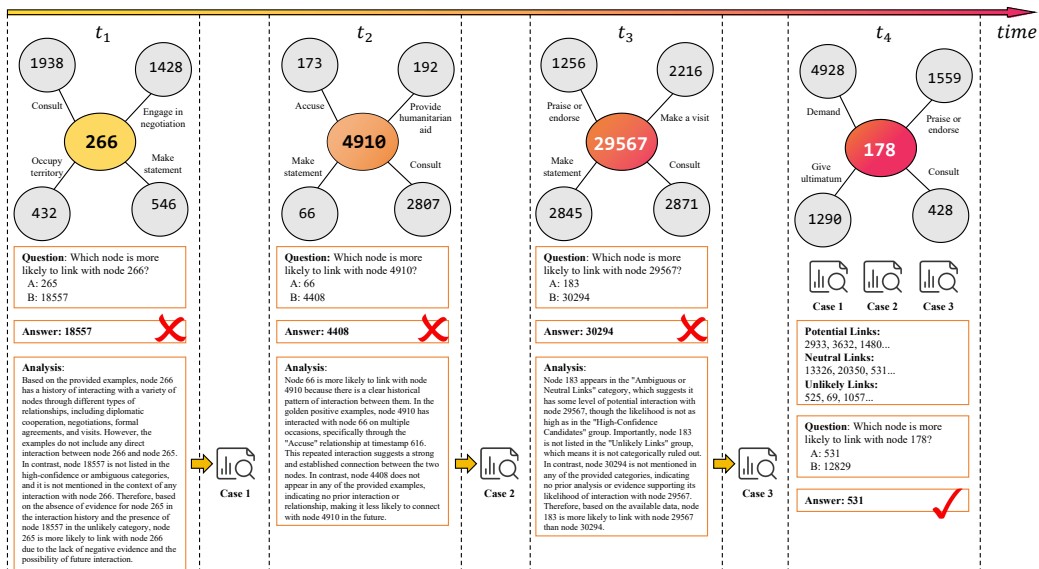

Figure 6: Case study for self-diagnosis reasoning. To prevent the LLM from having seen the test data during the training phase, we replace all entity names with unique identifiers. This process uses 3-shot examples, which demonstrate the LLM's process of discovering errors → error reflection → case-based reasoning during testing.

**Case 2 (using score-based tiers as explicit evidence).** For query node $u$=29567 (choices 183 vs. 30294), the model's reflective analysis cites our score-based partitioning shown in the prompt: node 183 appears in the *Ambiguous/Neutral* tier (hence not ruled out), while 30294 is absent from all tiers, indicating no supporting evidence; thus 183 is deemed more likely.[7] This illustrates how the module converts numerical scores into human-readable, verifiable cues that the LLM can reuse.

**Why it works with ID-only entities.** Even without textual semantics, the dynamic prompt exposes (i) **structural/historical regularities** through *Golden Positive* and time-stamped traces, and (ii) **uncertainty structure** through the three-way partition that surfaces high-confidence, ambiguous, and unlikely candidates. The reflective prompt then asks the model to *explain why the correct node was favored by the evidence*, and the resulting explanation is cached as a few-shot example (capacity-limited) and prepended for future queries—yielding a closed-loop improvement cycle at test time.

**Outlook.** Since the two cases above succeed *purely* from structural evidence and score-tier cues, we anticipate further gains by adding lightweight textual descriptors (entity names/types), which would sharpen self-similarity terms and reduce ambiguity within the *Ambiguous/Neutral* tier, while keeping the self-diagnosis mechanism unchanged.

# I    COMPLEXITY AND RUNTIME ANALYSIS

**Complexity of DyCo-LLM.** Let $d_u$ be the degree of the query node, $P = |P(u)|$ the number of historical partners, $K = |C|$ the final candidate pool size after multi-path recall, and $d_{sem}, d_{str}$ the embedding dimensions. Per query, adaptive hyperparameter computation uses constant-time statistics (e.g., $\log d_u$, $\Delta t_k$), hence $O(1)$. *Multi-path recall* costs $O(k_{time})$ for the time-active scan (implemented as a bounded backward sweep), $O(\tilde{k}_{struc})$ for structural recall using adjacency sketches or precomputed co-neighbor indices ($\tilde{k}_{struc} \leq k_{struc}$ after deduplication), and $O(k_{sem})$ for semantic recall with top-$k$ ANN over fixed entity embeddings. Thus recall is $O(k_{time}+\tilde{k}_{struc}+k_{sem}) = O(K)$.

---

[7]The prompt is organized into *Golden Positive*, *Potential Future*, *Ambiguous/Neutral*, and *Unlikely* blocks, which the LLM can reference during reflection.

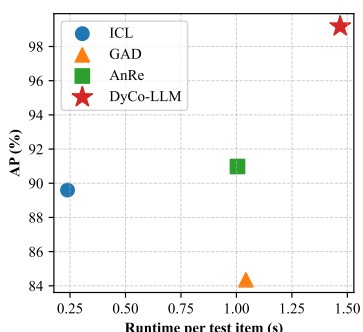

Figure 7: Comparison between model performance and per-item runtime.

*Scoring* forms similarities to $P$ partners in two spaces and aggregates with $W(\cdot)$: the matrix–vector accumulation is

$$O\Big(KP\,(d_{\text{sem}}+d_{\text{str}})\Big) \;+\; O\Big(K(d_{\text{sem}}+d_{\text{str}})\Big)$$

for history and self terms, respectively; thresholding and partitioning are $O(K \log K)$ (or $O(K)$ with selection). Therefore, the dominant per-query complexity is

$$O\Big(KP\,(d_{\text{sem}}+d_{\text{str}}) + K \log K\Big),$$

and the overall wall-clock is linear in the processed test events since updates are *streaming*. *In-situ updates* of the Random Projection (RP) module for a new event $(x, y, t)$ cost $O(L \cdot d_{\text{str}})$ (time-decayed $L$-hop projection with constant fan-out), and adjacency/history maintenance is $O(1)$. The reflective few-shot step is triggered sparsely (only on errors/low confidence); its amortized cost is bounded by a constant factor on top of the base LLM decoding because the few-shot buffer size is capped (e.g., 3 shots). In practice, $K$ and $P$ are *adaptively bounded* by $(k_{\text{time}}, k_{\text{struc}}, k_{\text{sem}})$ and the time-decayed history, so the effective compute scales sublinearly with the ambient graph size $|\mathcal{E}|$. This explains the favorable runtime we observe while preserving high recall and strong accuracy.

**Runtime–Accuracy Trade-off.** As shown in Figure 7, we further compare the per-item runtime against accuracy (AP) across test-time methods. Our **DyCo-LLM** attains the highest AP of **99.18** at 4.66 s/it. Although DyCo-LLM is slower than lightweight LLM baselines on a per-item basis, the *absolute* gain in AP (+9.58% over ICL and +8.20% over the strongest baseline AnRe) places the method at a favorable point on the runtime–accuracy frontier. Importantly, DyCo-LLM is *training-free*: it avoids any offline parameter learning or epoch-wise optimization required by graph neural models, whose wall-clock cost is dominated by multi-epoch backpropagation over temporal minibatches. In contrast, DyCo-LLM only performs bounded candidate recall and similarity aggregation at test time, with streaming state updates and a capped few-shot memory. Consequently, while our per-item inference is moderately heavier than prompt-only LLM variants, the *total* compute to deploy a high-accuracy system is substantially lower than training-based DGNN pipelines—no pretraining, no fine-tuning, and no re-training when the test distribution drifts. This property is critical for dynamic KGs where rapid adaptation outweighs amortized training efficiency.

## J   USE OF LLMs

This article employed LLMs to refine certain aspects of writing logic and grammatical accuracy. In the experimental code section, some portions of the code were generated with the assistance of LLMs. However, LLMs were not involved in the formulation of the core ideas or the overall structure of the manuscript.

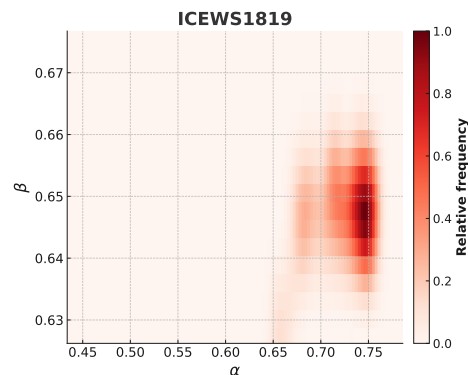 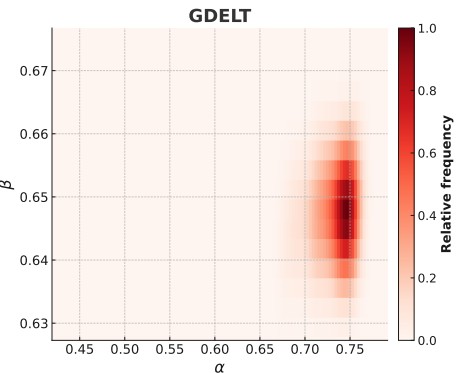

Figure 8: The relative frequency distribution of $(\alpha, \beta)$ on ICEWS1819 and GDELT.

Table 4: The formal definition of notations used throughout the paper.

| Symbol | Description |
|---|---|
| $\mathcal{E}, \mathcal{R}, \mathcal{T}$ | Entity / relation / timestamp sets |
| $G = \{(u, r, v, t)\}$ | Temporal knowledge graph as a set of timestamped facts |
| $G_{<t}$ | History up to (but excluding) time $t$ |
| $(u, r, ?, t)$ | Query quadruple (head $u$, relation $r$, unknown tail at time $t$) |
| $P(u)$ | Historical partners of $u$ before $t$ |
| $C$ | Candidate set returned by multi-path recall |
| $C_{\text{struc}}, C_{\text{sem}}, C_{\text{time}}$ | Structural / semantic / temporal recall subsets |
| $k_{\text{struc}}, k_{\text{sem}}, k_{\text{time}}$ | Adaptive recall budgets for the three paths |
| $\alpha, \beta$ | Adaptive mixing weights (structure vs. semantics; history vs. self) |
| $\mathbf{E}_{\text{sem}}(\cdot), \mathbf{E}_{\text{struc}}(\cdot)$ | Semantic / structural embeddings |
| $\text{sim}_{\text{sem}}, \text{sim}_{\text{str}}$ | Cosine similarity in semantic / structural space |
| $\mathbf{W}(p)$ | Historical partner weight for $p$: frequency $\times$ time-decay |
| $\lambda_{\text{time}}$ | Time-decay rate in $\mathbf{W}(p)$ |
| $S_{\text{hist}}(v), S_{\text{self}}(v)$ | Historical similarity and self-similarity scores for $v$ |
| $S_{\text{final}}(v)$ | Final score used for ranking candidates |
| $p, q$ | Thresholds (or quantiles) for partitioning $C$ into pos/amb/neg sets |
| $d_u, n_u$ | Degree and number of past interactions of node $u$ |
| $\mu_{\log \deg}, \mu_{\Delta t}, \mu_\sigma$ | Global statistics: avg. log-degree, inter-event gap, similarity std |
| $\Delta t_k$ | Gap to the $k$-th most recent event before $t$ |
| $L$ | Max hop (projection depth) in Random Projection module |
| $d_{\text{sem}}, d_{\text{str}}$ | Dimensions of semantic / structural embeddings |
| $K = |C|$ | Total number of recalled candidates |
| $\text{AP}, \text{AUC}, \text{MRR}, \text{Hits@K}$ | Evaluation metrics used in experiments |

| Section | Template |
|---|---|
| **Preamble** | You are a link prediction expert in a dynamic graph. Based on the examples of past interactions, determine which of the two new nodes is more likely to connect with the query node. In the following quadruple '(u, r, v, t)' examples, 'u' is the source node ID, 'r' is the text describing the link type, 'v' is the destination node ID, and 't' is the timestamp of the interaction. |
| **Expert Reasoning Examples** | Here are some previous cases with expert analysis. Learn from them:
--- Example Start ---
Prompt Context:
{failed_prompt_template}
Expert Analysis:
{reasoning_text}
--- Example End ---
... |
| **Golden Positive Examples** | ### Golden Positive Examples (Confirmed Past Interactions)
These links have definitely happened:
({u_id}, {r_his}, {v_his}, {t_his}) ... **(one per line)** |
| **Potential Future Links** | ### Potential Future Links (High-Confidence Candidates)
Based on analysis, these links are highly likely to happen in the future:
({u_id}, might interact with, {v_future}, {t_query}) ... |
| **Ambiguous / Neutral Links** | ### Ambiguous or Neutral Links (Uncertain Candidates)
Based on analysis, the likelihood of these links happening is uncertain:
({u_id}, might interact with, {v_neutral}, {t_query}) ... |
| **Unlikely Links** | ### Unlikely Links (Low-Confidence Candidates)
Based on analysis, these links are very unlikely to happen:
({u_id}, interact with, {v_unlikely}, {t_query}) ... |
| **Instruction & Question** | Now, based on all the examples above, analyze the examples and answer the following question. The correct answer could be either A or B. You must only output the letter of the correct option A or B.
Question: Which node is more likely to link with node {u_id}?
A: {v_true}
B: {v_neg}
Answer: |

Figure 9: The prompt template for structured reasoning.

| Section | Template |
|---|---|
| **Preamble** | You are a link prediction expert in a dynamic graph. Based on the examples of past interactions, determine which of the two new nodes is more likely to connect with the query node. In the following quadruple '(u, r, v, t)' examples, 'u' is the source node ID, 'r' is the text describing the link type, 'v' is the destination node ID, and 't' is the timestamp of the interaction. |
| **Golden Positive Examples** | ### Golden Positive Examples (Confirmed Past Interactions)
These links have definitely happened:
({u_id}, {r_his}, {v_his}, {t_his}) ... **(one per line)** |
| **Potential Future Links** | ### Potential Future Links (High-Confidence Candidates)
Based on analysis, these links are highly likely to happen in the future:
({u_id}, might interact with, {v_future}, {t_query}) ... |
| **Ambiguous / Neutral Links** | ### Ambiguous or Neutral Links (Uncertain Candidates)
Based on analysis, the likelihood of these links happening is uncertain:
({u_id}, might interact with, {v_neutral}, {t_query}) ... |
| **Unlikely Links** | ### Unlikely Links (Low-Confidence Candidates)
Based on analysis, these links are very unlikely to happen:
({u_id}, interact with, {v_unlikely}, {t_query}) ... |
| **Original Question** | Now, based on all the examples above, analyze the examples and answer the following question. The correct answer could be either A or B. You must only output the letter of the correct option A or B.
Question: Which node is more likely to link with node {u_id}?
A: {v_true}
B: {v_neg} |
| **Case Analysis** | The correct answer was A: {v_true}. Based on the context provided in the prompt, please provide a brief analysis explaining why node {v_true} was the more likely connection.
Reasoning: |

Figure 10: The prompt template for few-shot example construction.

