# OpenReview forum: "In-situ Adaptation for LLM-based Link Prediction: A Dynamic Cognition Paradigm for Temporal Knowledge Graphs"
_ICLR.cc/2026/Conference — ICLR 2026 Conference Withdrawn Submission_

### Official Review · Reviewer_UWpm · 2025-10-16

**Soundness:** 1
**Presentation:** 3
**Contribution:** 1
**Rating:** 2
**Confidence:** 5

**Summary:**

This paper proposes DyCo-LLM, a training-free framework for temporal knowledge graph (TKG) forecasting/ future link prediction on TKG. The method creates LLM prompts on the fly.
It comes with an adaptive multi-path recall and scoring mechanism that adjusts its parameters based on evolving node and graph-level features.
Experiments are conducted on the DTGB benchmark (NeurIPS 2024) using datasets such as ICEWS1819 and GDELT. The paper reports strong results under AUC and Average Precision metrics.

**Strengths:**

* The general idea of test-time, training-free adaptation for LLM reasoning is interesting and could inspire future research on dynamic or streaming environments.
* The system architecture is clearly presented and visually well explained.
* Writing quality and presentation are strong.

**Weaknesses:**

1. Mismatch: Future Link prediction/ Forecasting for Temporal Knowledge Graphs vs dynamic text-attributed graphs
* The paper claims to be a contribution to temporal knowledge graph forecasting (Future Link prediction for temporal knowledge graphs)
* But: The actual setup follows the DTGB benchmark, which models dynamic text-attributed graphs, mostly single-relation graphs without explicit relation reasoning.
* DyCo-LLM does not model or even use relation types (besides the final prompt), which are central to TKG reasoning.
* The paper ignores the complete body of work on TKG Forecasting.
* In my opinion, for these reasons, the work is conceptually misaligned with the TKG domain it claims to advance.

2. Evaluation Setup Incompatible with TKG Standards
* The paper uses DTGB’s [7] evaluation protocol. This frames link prediction as a binary edge classification task and reports AUC and Average Precision (AP). While this is consistent with DTGB, it diverges fundamentally from how temporal knowledge graph forecasting is evaluated in the literature. For reference, see e.g. [1]
* In TKG forecasting, the model is expected to assign plausibility scores to candidate entities for a query
(s,r,?,t), and performance is assessed by how highly the correct entity ranks among all possible candidates. Metrics such as (time aware filtered) Mean Reciprocal Rank (MRR) and Hits@K therefore measure whether the model understands relational and temporal structure well enough to identify the true target among all options.
* By contrast, the binary setup used here only requires the model to distinguish the correct entity from one (mostly randomly chosen) negative. This makes the task much easier and does not test fine-grained ranking ability or temporal reasoning.
The near-perfect AUC/AP scores reported in Table 1 confirm that the evaluation is not meaningfull.
* As a consequence, the reported results cannot be compared to prior TKG forecasting work (e.g., RE-NET, RE-GCN, TLogic, TIRGN, CognTKE, Recurrency Baseline), all of which follow the ranking-based setup.
* Using AUC and AP also invalidates claims of “state-of-the-art” performance within the TKG forecasting domain.

3. Inappropriate and Incomplete Baselines
* The baselines (TGAT, DyRep, JODIE, ICL, AnRe, GAD) are not TKG forecasting methods but for single-relational graphs.
* No comparison is made to actual TKG baselines, making it impossible to judge whether DyCo-LLM advances the field.
* I recommend comparing to (at least!) the following: RE-GCN [2], TLogic [5], TIRGN [3], CognTKE [4], Recurrency Baseline [6].
* Datasets: Typically, for TKG forecasting, the datasets ICEWS14,ICEWS18,GDELT,WIKI,YAGO (see [1]) or potentially from TGB 2.0 benchmark are used. I strongly recommend to include them in your experiments.

4. Conceptual and Modeling Issues
* It is not clear, why you propose the self-similarity score between head and tail entities. An example: for the quadruple [Obama, visits, France, 2014-01], Obama is not at all semantically similar to France. This is true for many (most?) quadruples.
Further, semantic similarity is computed without conditioning on the relation.
* Temporal recall similarly ignores relation type (retrieves recent nodes globally).
* Together these suggest that the method has been adapted from a general dynamic-graph context without careful consideration of what makes TKG reasoning distinctive.
* The so-called “hyperparameters” are actually runtime-computed variables (based on some predefined heuristics and dataset stats), not true hyperparameters.

5. Missing Related Work
* The related work section does not contain most (or any) key contributions in TKG forecasting, including RE-NET, RE-GCN, TLogic, TIRGN, CognTKE, Recurrency Baseline, and many more.
* This again gives the feeling that the authors do not know the TKG domain, and the paper is a Temporal single-relational instead of TKG paper.

6. Overstated Claims and Framing
* The paper repeatedly claims to “unify dynamic graph learning and LLM memory research” and achieve “state-of-the-art” results
* But, the method mainly combines heuristic adaptation rules with existing pretrained components. The “Dynamic Cognition” framing is very rhetorical, the method does not demonstrate new cognitive or algorithmic capabilities.

7. Potential Test-Set Leakage via Pretrained LLM
* The method uses a pretrained LLM (Qwen3-8B) on datasets containing real-world events from e.g. 2018–2019 (ICEWS1819).
* Since the LLM was trained on data likely covering the same period, it is likely that many test events were already seen during pretraining.
* This leads to a serious data leakage risk: The model might recall facts directly rather than infer them.
* No steps are described to mitigate or verify against such leakage.

## Summary

In summary, while DyCo-LLM presents an interesting idea of test-time adaptation for LLMs on dynamic graphs, the paper’s evaluation framework, task formulation, and baselines are misaligned with temporal knowledge graph forecasting standards.
By adopting the DTGB binary-classification setup instead of the established ranking-based MRR/Hits@K evaluation, the work does not actually test TKG reasoning.
Combined with conceptual issues, missing related work, and potential test-set leakage from pretrained LLMs, the submission lacks the rigor and comparability required for acceptance at ICLR.

## References

[1] Gastinger, J., Sztyler, T., Sharma, L., Schuelke, A., & Stuckenschmidt, H. (2023, September). Comparing apples and oranges? on the evaluation of methods for temporal knowledge graph forecasting. In Joint European conference on machine learning and knowledge discovery in databases (pp. 533-549). Cham: Springer Nature Switzerland.

[2] Zixuan Li, Xiaolong Jin, Wei Li, Saiping Guan, Jiafeng Guo, Huawei Shen, Yuanzhuo Wang, and Xueqi Cheng. Temporal knowledge graph reasoning based on evolutional representation learning. In The 44th International ACM SIGIR Conference on Research and Development in Information Retrieval (SIGIR), 2021b.

[3] Yujia Li, Shiliang Sun, and Jing Zhao. TiRGN: Time-guided recurrent graph network with local- global historical patterns for temporal knowledge graph reasoning. In Proceedings of the 31st International Joint Conference on Artificial Intelligence (IJCAI), pp. 2152–2158, 2022a.

[4] Wei Chen, Yuting Wu, Shuhan Wu, Zhiyu Zhang, Mengqi Liao, Youfang Lin, and Huaiyu Wan. Cogntke: A cognitive temporal knowledge extrapolation framework. In Proceedings of the AAAI Conference on Artificial Intelligence, volume 39, pp. 14815–14823, 2025.

[5] Yushan Liu, Yunpu Ma, Marcel Hildebrandt, Mitchell Joblin, and Volker Tresp. TLogic: Temporal logical rules for explainable link forecasting on temporal knowledge graphs. In 36th Conference on Artificial Intelligence (AAAI), pp. 4120–4127, 2022

[6] Julia Gastinger, Christian Meilicke, Federico Errica, Timo Sztyler, Anett Schuelke, and Heiner Stuckenschmidt. History repeats itself: A baseline for temporal knowledge graph forecasting. In Proceedings of the Thirty-Third International Joint Conference on Artificial Intelligence (IJ- CAI), 2024b

[7] Zhang, J., Chen, J., Yang, M., Feng, A., Liang, S., Shao, J., & Ying, R. (2024). DTGB: A comprehensive benchmark for dynamic text-attributed graphs. Advances in Neural Information Processing Systems, 37, 91405-91429.

**Questions:**

1. How would DyCo-LLM perform under ranking-based metrics (time-aware filtered MRR, Hits@K) in a true TKG forecasting setup?

2. How is the relation type (r) besides the prompt to the LLM, if at all?

3. Why was DTGB chosen as the main benchmark given its task mismatch with TKG forecasting?

4. What measures did you take to prevent test-set leakage from pretrained LLMs on datasets covering 2018–2019 events?

---

### Official Review · Reviewer_ULjv · 2025-10-23

**Soundness:** 1
**Presentation:** 3
**Contribution:** 2
**Rating:** 2
**Confidence:** 4

**Summary:**

The authors introduce DyCo-LLM, a training-free framework enabling LLMs to perform real-time, in-situ adaptation for TKG link prediction. DyCo-LLM continuously adjusts its inference behavior through a closed-loop system comprising adaptive runtime diagnostics, multi-path recall (structural, semantic, and temporal), dynamic score fusion, and self-diagnosis reasoning that converts prediction errors into corrective few-shot examples. Experiments on two TKG dataset ICEWS18–19 and GDELT demonstrate that the proposed method can outperform dynamic GNNs and LLM methods.

**Strengths:**

- **good presentation**: the paper presentation is clear and easy to follow. The diagrams and tables are also well-presented
- **query time adaption**: the authors proposed a novel method for query-level adaptation via runtime diagnosis, achieving in-situ adaptation without updating weights.
- **code provided**: code is provided with the submission for reproducibility.

**Weaknesses:**

The main limitation of this work is its empirical evaluation, the following three aspects are limited.

- **lack of dataset diversity**: the authors only benchmarked on two TKG datasets and on these datasets: ICEWS1819 and GDELT, most methods have achieved > 95% performance based on the AP and AUC-ROC, further showing that the datasets are too easy for evaluation and more diverse benchmark is needed.
- **evaluation metric should be a ranking metric**. Evaluation on TKGs are often evaluated with a ranking metric such as MRR or Hits@K. These metrics are commonly used in prior TKG literature and often significantly more challenging than the AP / AUC-ROC metric where they treat the evaluation as a binary classification. You can see MRR or Hits@K metric in TLogic[1], 	Recurrency Baseline[2] and RE-NET[3]. All these paper also evaluated on 4-5 datasets.
- **lack of TKG baselines**. As the method is proposed for temporal knowledge graph, it should be compared directly with SOTA TKG methods. In the paper however, the authors focused on Dynamic GNN and LLM methods only but overlooking the TKG methods. TKG methods such as TLogic and RE-Net was cited but not compared.

Due to the above issues with the empirical experiments, the paper didn't provide me with sufficient evidence to see the actual performance of DyCo-LLM. My suggestions are as follows:

- **test on more datasets**. Testing on at least 4-5 datasets can strengthen the conclusion from the paper, you can find datasets from the references below. Recently, TGB 2.0[4] was proposed to incorporate large scale and challenging TKG datasets as well, would be interesting to evaluate the scalability of the proposed method as well.

- **evaluation metric should be a ranking metric**. Either use MRR or Hits@K for better and more robust evaluation where the true edge is ranked against multiple negatives

- **adding TKG baselines**. Add baselines designed specifically for TKG such as TLogic, RE-NET and Recurrency Baseline.

[1] Liu Y, Ma Y, Hildebrandt M, Joblin M, Tresp V. Tlogic: Temporal logical rules for explainable link forecasting on temporal knowledge graphs. InProceedings of the AAAI conference on artificial intelligence 2022 Jun 28 (Vol. 36, No. 4, pp. 4120-4127).

[2] Gastinger J, Meilicke C, Errica F, Sztyler T, Schuelke A, Stuckenschmidt H. History repeats itself: A baseline for temporal knowledge graph forecasting. arXiv preprint arXiv:2404.16726. 2024 Apr 25.

[3] Jin W, Qu M, Jin X, Ren X. Recurrent event network: Autoregressive structure inference over temporal knowledge graphs. arXiv preprint arXiv:1904.05530. 2019 Apr 11.

[4] Gastinger J, Huang S, Galkin M, Loghmani E, Parviz A, Poursafaei F, Danovitch J, Rossi E, Koutis I, Stuckenschmidt H, Rabbany R. Tgb 2.0: A benchmark for learning on temporal knowledge graphs and heterogeneous graphs. Advances in neural information processing system

**Questions:**

- line 159, $k_{str}$, $k_{sem}$, $k_{time}$, $\alpha$ and $\beta$ notations were never introduced there and only explained much later in line 172 and later
- how would semantic recall work on standard temporal graphs without text-features?
- what does this sentence mean? "fixed windows or hierarchical paging cannot expand/update with event accumulation, long-context recall amplifies noise" line 52

**Details Of Ethics Concerns:**

No Ethical concerns

---

### Official Review · Reviewer_HGgg · 2025-10-28

**Soundness:** 1
**Presentation:** 1
**Contribution:** 1
**Rating:** 0
**Confidence:** 5

**Summary:**

This paper introduces a training-free procedure to perform link-prediction forecasting on temporal knowledge graphs (TKGs). The overall idea is to have an LLM answer a query based on 5 parameters that depend on the characteristic of the TKG. These parameters inform other modules, which compute several more heuristics eventually conditioning the answer of the LLM. The evaluation is performed on two TKG forecasting benchmarks.

**Strengths:**

We must acknowledge the merit of the paper in informing the decision process of heuristics that are inspired by the temporality of the task as well as the structural characteristics of the graph.

**Weaknesses:**

It is my belief that this paper has large margins of improvement, starting  from how the paper has been written. The text is extremely verbose, relying on the use of adjectives that are often unnecessary or inadequate as it often happens when writing a paper using LLMs. The authors acknowledge the use of an LLM to aid the writing, but it feels as if the paper had been heavily written with it, which is not wrong per se but in this specific case carries a number of problems with it. Besides being hard to follow, despite the simplicity of the approach, it is unclear what it is that the authors strive to address (lines 53-54). The introduction looks like a related work section with the addition of a description of the proposed system that is completely obscure to the reader at that point in time. The use of words as “in-situ” is inconsistently used: sometimes they refer to the actual meaning, other times (line 221) their use makes little sense. It is unclear why Figure 1 is so important for the understanding of the proposed methodology, since ICL does not seem to appear explicitly in other parts of the text after Section 2. Images are unclear and possibly some symbols are broken (there is a “?” which overlaps with node indices, which makes me think there should have been another symbol there). Subscripts are used inconsistently, for example $k_{str}$ and $k_{struc}$. The organization of the paper could be improved, it is made of too many titled paragraphs which do not help with the flow and the Adaptive Parameters section, which is the first step of the pipeline, is explained at the very end of Section 3. Overall, in terms of clarity of presentation, the paper is insufficient for a top-tier conference and needs major improvements. I would suggest that the authors simplify the text by reducing the verbosity and improving consistent use of certain terms, in addition to explaining more clearly the specific problem being addressed by your solution and why it is important. Clear intuitions should be also added while explaining the methodology.

In terms of experiments, I disagree that the standard metrics are AP and AUC-ROC. Most reproducible works on TKG forecasting use MRR and Hits@10, especially on the two datasets the authors benchmark their methods against. I would suggest that the authors report MRR and Hits@10 scores instead. If the authors disagree with my interpretation, I would appreciate it if they could provide references where AP and AUC-ROC are used, and I will provide alternative references instead to support my argument.

The evaluation is limited, especially considering that the approach proposed by the authors is training-free. Hyper-parameter tuning of the $\gamma$ terms is not discussed in the main paper, whereas it is important to understand how the authors chose these terms. Right now, we cannot know whether these terms were chosen to maximize test performances, which would be a severe violation of ML evaluation practices.

Overall, I think the paper would benefit by a clearer description of the evaluation process as well as additional experiments as in most TKG forecasting works.

**Questions:**

- Could you explain in detail how you performed hyper-parameter tuning of terms like the $\gamma_{struc}$?

---

### Official Review · Reviewer_w2S8 · 2025-11-01

**Soundness:** 3
**Presentation:** 3
**Contribution:** 3
**Rating:** 6
**Confidence:** 4

**Summary:**

This paper introduces DyCo-LLM, a training-free and test-time adaptive framework for temporal knowledge graph (TKG) link prediction. The key idea is to treat the LLM as a self-regulating cognitive agent capable of diagnosing each query’s local context and adapting its reasoning dynamically. A dynamic context engine adjusts recall, scoring, and prompting in real time through four modules. Experiments on ICEWS and GDELT show that DyCo-LLM achieves state-of-the-art results among both parameter-trained and LLM-based baselines, demonstrating inductive generalization without any retraining.

**Strengths:**

- The paper presents a novel, training-free paradigm for temporal reasoning, which is both conceptually interesting and practically relevant for dynamic environments.
- The closed-loop adaptive design is well motivated.
- Writing quality and methodological descriptions are generally clear and well structured.

**Weaknesses:**

- The authors mention using Qwen3-8B as the backbone, but it remains unclear whether the same model size and configuration are used for all LLM-based baselines, which could affect the fairness of the comparison.
- How sensitive is DyCo-LLM to the base LLM’s size and context window?

**Questions:**

See weaknesses.

---

### Official Review · Reviewer_zdCN · 2025-11-01

**Soundness:** 2
**Presentation:** 2
**Contribution:** 2
**Rating:** 2
**Confidence:** 5

**Summary:**

The paper presents DyCo-LLM, a method that enables an LLM to perform temporal link prediction.

**Strengths:**

1.	Addresses an important and timely problem in temporal dynamic graphs.

**Weaknesses:**

1.	The proposed link-prediction approach does not align well with the paper’s stated research objectives.

**Questions:**

1.	The problem formulation is unclear. The introduction claims to tackle two key aspects of dynamic graphs, i.e. continual accumulation of events and shifting relevance of historical information, but the method focuses narrowly on temporal link prediction and does not clearly propose techniques that handle continual data accumulation or time-varying relevance.
2.	The paper assumes a preconstructed temporal knowledge graph. In practice, building high-quality temporal KGs from domain text (e.g., biomedicine or finance) is a major challenge. Given the stated goal of in-situ adaptation for dynamic graphs, the work should clarify its scope or include methods for dynamic data management and KG construction/maintenance.

---

### Note · Authors · 2025-11-20

I have read and agree with the venue's withdrawal policy on behalf of myself and my co-authors.